# Diaphragm Fatigue in SMNΔ7 Mice and Its Molecular Determinants: An Underestimated Issue

**DOI:** 10.3390/ijms241914953

**Published:** 2023-10-06

**Authors:** Francesca Cadile, Deborah Recchia, Massimiliano Ansaldo, Paola Rossi, Giorgia Rastelli, Simona Boncompagni, Lorenza Brocca, Maria Antonietta Pellegrino, Monica Canepari

**Affiliations:** 1Department of Molecular Medicine, via Forlanini 6, University of Pavia, 27100 Pavia, Italy; francesca.cadile01@universitadipavia.it (F.C.); massimiliano.ansaldo01@universitadipavia.it (M.A.); lorenza.brocca@unipv.it (L.B.); map@unipv.it (M.A.P.); 2Department of Biology and Biotechnology “L. Spallanzani”, University of Pavia, 27100 Pavia, Italy; deborah.recchia@unipv.it (D.R.); paola.rossi@unipv.it (P.R.); 3Center for Advanced Studies and Technology, University G. d’Annunzio of Chieti-Pescara, 66100 Chieti, Italy; giorgia.rastelli@unich.it (G.R.); simona.boncompagni@unich.it (S.B.); 4Department of Neuroscience, Imaging and Clinical Sciences, University G. d’Annunzio of Chieti-Pescara, 66100 Chieti, Italy

**Keywords:** SMA, diaphragm muscle, muscle fatigue, ergothioneine

## Abstract

Spinal muscular atrophy (SMA) is a genetic disorder characterized by the loss of spinal motor neurons leading to muscle weakness and respiratory failure. Mitochondrial dysfunctions are found in the skeletal muscle of patients with SMA. For obvious ethical reasons, the diaphragm muscle is poorly studied, notwithstanding the very important role that respiratory involvement plays in SMA mortality. The main goal of this study was to investigate diaphragm functionality and the underlying molecular adaptations in SMNΔ7 mice, a mouse model that exhibits symptoms similar to that of patients with intermediate type II SMA. Functional, biochemical, and molecular analyses on isolated diaphragm were performed. The obtained results suggest the presence of an intrinsic energetic imbalance associated with mitochondrial dysfunction and a significant accumulation of reactive oxygen species (ROS). In turn, ROS accumulation can affect muscle fatigue, cause diaphragm wasting, and, in the long run, respiratory failure in SMNΔ7 mice. Exposure to the antioxidant molecule ergothioneine leads to the functional recovery of the diaphragm, confirming the presence of mitochondrial impairment and redox imbalance. These findings suggest the possibility of carrying out a dietary supplementation in SMNΔ7 mice to preserve their diaphragm function and increase their lifespan.

## 1. Introduction

SMA is an autosomal recessive genetic disorder characterized by loss of spinal motor neurons leading to muscle atrophy, weakness, and respiratory failure [1]. SMA has an incidence of about one in 10,000 live births [2]. It is caused by mutations in the SMN1 gene, which encodes the Survival Motor Neuron (SMN) protein, essential for the survival and normal functioning of motor neurons [3].

In humans, there are two SMN genes, the telomeric SMN1 coding for a ubiquitous protein (full-length SMN), and its centromeric homolog SMN2 mostly generating a protein lacking exon7 (SMNΔ7), which is not functional: the SMN2 gene produces a limited amount of functional protein that can modulate SMA severity [4].

Although the key role of motoneurons in determining the most severe symptoms of SMA is widely recognized, the presence of defects in other cell/tissue types in SMA models indicates the complexity of this disease [5,6,7]. Pathological changes in the SMA skeletal muscle system have been widely reported, and a direct contributing role of muscle in SMA was also hypothesized [8,9]. Besides denervation, available data also indicate the presence of mitochondrial disturbances in the locomotor muscles of patients and mouse models of SMA [10,11,12].

The effect of SMA on the skeletal muscle system is not uniform, and the muscle groups that are affected by the disease and those spared are similar in both humans and mouse models of SMA [13]. Respiratory involvement plays an important role in morbidity and mortality in children [14] and mice [15] with SMA. It is, in fact, necessary to monitor respiratory muscle performance in order to avoid acute and/or chronic respiratory failure [16]. Despite the relevance of this aspect, for obvious ethical reasons, very few studies have characterized the diaphragm muscle in patients with SMA. Specifically, based on clinical evaluations, relatively well-preserved diaphragm strength was found in patients with intermediate type II SMA as well as an increase in fatigue sensitivity that eventually leads to acute respiratory failure, especially during sleep [16]. Very few studies have analyzed the molecular level of the human SMA diaphragm. Degenerative changes in neuromuscular junctions (NMJs) in the diaphragm have been described in 6-month-old patients [17], and no clear damage was found in one post-mortem diaphragm muscle sample from a patient with SMA type I [10].

Several murine models of SMA have been generated, mostly based on Smn1 (murine homolog of human SMN1) gene inactivation and the introduction of more copies of the human SMN2. Severe SMA mouse models exhibit defects in breathing patterns with smaller ventilation volume, longer breath duration, and greater apnea frequency and duration [15]. Moreover, pronounced swelling of mitochondria [18] and alteration in the expression of proteins with an important role in the mitochondrial respiratory chain were found at the level of sub-synaptic regions [19].

The relevance of respiratory muscles to the clinical picture of patients with SMA and the lack of data on the intrinsic contractile capacity of the diaphragm warrant further investigations. An understanding of the diaphragm muscle defects could be important for preventing and effectively managing respiratory complications in SMA patients.

The central goal of this study was to carry out a comprehensive analysis of the intrinsic diaphragm functionality and the underlying molecular adaptations, combining functional, biochemical, and molecular analyses on isolated diaphragm muscle of SMNΔ7 mice. This model is, to date, one of the lines of SMA mice most widely used in numerous laboratories around the world. Ref. [20] exhibited symptoms and neuromuscular pathology similar to that of patients with intermediate type II SMA [21,22]. In this mouse model of SMA, ultrastructural study revealed features of NMJ alterations in diaphragm muscle [18], but no information about its contractile function is available. The results obtained in this study showed high fatigue sensitivity related to a redox imbalance in the diaphragm from SMNΔ7 mice. Interestingly, exposure of the diaphragm muscle to ergothioneine, a potent antioxidant molecule, seems able to restore diaphragm function.

## 2. Results

### 2.1. Ex Vivo Functional Analysis

The contractile performance was assessed on diaphragm muscles isolated from SMNΔ7 (SMA) and control (WT) mice. Twitch time parameters, i.e., time to peak (TTP) and half relaxation time (TTR), showed a significant prolongation in the SMA diaphragm. Consequently, the isometric specific tension developed in twitch was increased, while specific tetanic tension was not changed (Figure 1a). Importantly, the fatigue test showed that the fatigue resistance was significantly reduced in SMA diaphragm in comparison with WT (Figure 1b).

The kinetics of force development and relaxation are mainly dictated by Ca2+ handling in the cell. A significant decrease in sarcoplasmic reticulum (SR) Ca2+ ATPase (SERCA1 and 2) pump gene and protein expression was found (Figure 1c).

### 2.2. Energy Imbalance and Oxidative Metabolism

A significantly increased phosphorylation of the energy sensor AMPK was found in the SMA diaphragm (Figure 2a), suggesting a higher AMP/ATP ratio in the muscle and the presence of energy imbalance. Gene expression of PGC1α, Sirtuin1, and NRF1, involved in the regulation of oxidative metabolism by stimulating mitochondrial biogenesis, and the PGC1α protein expression were found significantly downregulated in the SMA diaphragm (Figure 2b,c). Unexpectedly, however, the protein expression of mitochondrial import receptor subunit TOM 20 was significantly higher (Figure 2c) in the SMA diaphragm, suggesting an increased mitochondrial mass. In agreement, the key components of the mitochondrial respiratory chain (OXPHOS complex), in particular, complex I, III, and IV (Figure 3), were significantly up-regulated. To evaluate whether the increased level of mitochondrial complexes leads to enhanced mitochondrial function, we compared the respiratory capacity of the diaphragms of SMA and WT mice. No changes were found in leak cellular respiration, oxidative phosphorylation (OXPHOS), and electron transfer system capacity (ETS) sustained by complex I, complex II, and by the contribution of both (Figure 4).

To better characterize mitochondrial adaptations in the diaphragm of SMA mice, markers of mitochondrial dynamics were analyzed. Expression of pro-fusion proteins (OPA1, MNF1, MNF2), proteins involved in mitochondrial fission (FIS1, and the phosphorylate form of DRP1 at serine 616) showed no changes except for a significant increase in phosphorylation at serine 637 in DPR1 (Figure 5a,b).

### 2.3. Autophagy and Mitophagy

To assess the activation of the autophagy process, the ratio between the active and the inactive forms of LC3B (LC3BII/LC3BI) and the protein level of p62 were determined. The significant induction of the active form LC3BII, together with the decrease in p62 expression (Figure 6a), suggest the activation of the autophagy flux, which, however, does not result in an increased mitophagy activation. Indeed, a decrease in gene expression of BNIP3 (Figure 6b) and of PARKIN protein content, both involved in promoting the selective autophagy of depolarized mitochondria, was found (Figure 6c).

### 2.4. Redox Imbalance

Redox imbalance was investigated by evaluating the level of protein carbonylation, the level of ROS scavenger protein superoxide dismutase 1 (SOD1), catalase, Heat Shock Protein 27 (HSP27), Heat Shock Protein 70 (HSP70), Peroxiredoxin 3 (PRDX3) and gene expression of NRF2, the major sensor of cell redox balance. A significant increase in carbonylated proteins in the SMA diaphragm was revealed by Oxyblot analysis (Figure 7a). No induction of NRF2 (Figure 7b) and no changes in SOD1, catalase, HSP27, and HSP70 content were found. Peroxiredoxin 3 (PRDX3) was the only protein of antioxidant defense systems that revealed a significant increase in its expression (Figure 7c).

### 2.5. Effect of Exposure to the Antioxidant Molecule Ergothioneine

To test the impact of redox imbalance on contractile function, the determination of the contractile performance was repeated after incubation of the diaphragm in Krebs (for 30 min) and with 0.4% of water solution of ergothioneine (ERGO) added. Importantly, the exposure to ERGO restored TTP and TTR values to the WT level (Figure 8a) and reduced SMA diaphragm fatigability (Figure 8b).

## 3. Discussion

### 3.1. SMA Diaphragm Exhibits Reduced Fatigue Resistance

The lack of variation in specific tetanic tension (Figure 1a) suggests the function of the myofibrillar contractile machinery is preserved in the SMA diaphragm. This is in agreement with studies in patients with SMA demonstrating that the diaphragm function is quite spared [16]. Notwithstanding the capacity to develop tension, a significant increase in both time to peak tension and time to half relaxation of single twitch in diaphragms from SMNΔ7 mice was found. Kinetics of force development and relaxation are mainly dictated by Ca^2+^ handling in the cell. The rate of release of calcium ions by the sarcoplasmic reticulum (SR) depends on the activity of dihydropyridine and ryanodine receptors, whereas the rate of calcium reuptake is modulated by the activity of the SR calcium ATPase (SERCA) pumps. These pumps are particularly prone to suffer activity alterations, since they are characterized by an average life of 30–40 days, making these proteins potentially susceptible to oxidative damage [23]. In the present study, a significant reduction in gene and protein expression of SERCA1 and SERCA2 was found (Figure 1c). A very interesting result was the finding of a higher fatigability in diaphragm isolated from SMA mice (Figure 1b). This result is in accordance with the paper of Fauroux and colleagues suggesting that, in SMA patients, the inspiration relies mainly on the diaphragm that maintains its capacity to develop force but increases its fatigue sensitivity [16]. Therefore, our attention was focused on molecular determinants of diaphragm fatigue.

### 3.2. Energy Balance in SMA Diaphragm

One of the multiple factors leading to muscle fatigue could be a decrease in available ATP [24], as suggested by the finding of the activation of the energy sensor AMPK (Figure 2a). A decrease in available ATP could act on SERCA pumps and account for a longer timing of the single twitch. Moreover, a decrease in available ATP and a metabolic impairment could explain the greater sensitivity to fatigue observed in the diaphragm of SMA mice. The main source of ATP in the muscle is the oxidative phosphorylation that takes place in mitochondria. Our results showed that PGC1α, Sirtuin1, and NRF1 were significantly reduced in SMNΔ7 diaphragm, suggesting a defect in mitochondrial biogenesis (Figure 2b,c). Several indications of mitochondrial defects were found in the tibialis anterior muscle from SMNΔ7 mice [25,26] and muscle-specific Smn1 knockout mice [12], and in several skeletal muscles but not the diaphragm in patients with SMA [10]. Notice that only one diaphragm post-mortem was analyzed in these patients [10]. Interestingly, Chemello and colleagues demonstrated, in the tibialis anterior muscle of muscle-specific Smn1 knockout mice, the presence of dysfunctional mitochondria together with several unaltered mitochondria and higher mitochondrial protein levels [12]. Recently, it has been reported that mitochondrial activity was increased in an SMN depletion cell model [27]. In our study, an increased expression of mitochondrial import receptor subunit TOM-20 and marker proteins of OXPHOS complexes (Figure 2c and Figure 3), but no changes in basal mitochondrial activity, were found (Figure 4). These apparently conflicting results point to the presence of dysfunctional mitochondria that could account for the increase in mitochondrial proteins but not contribute to an increase in mitochondrial activity.

To ensure optimal function in energy generation, the dynamic properties of mitochondria are critical [28]. Among the mitochondrial-shaping proteins, the contribution of active dynamin-related protein 1 (DRP1) is one of the important factors affecting mitochondrial shape controls, calcium homeostasis, and muscle mass [29], and the phosphorylation plays a crucial role in its activity regulation [30]. Moreover, available evidence indicates that phosphorylation of DRP1 in Ser637 induces mitochondrial elongation against oxidative stress [31,32]. In the present study, a significant increase in Ser637 phosphorylation of DRP1 was found (Figure 5b), suggesting an alteration in mitochondrial shape. Fused mitochondria have been demonstrated in aging [33,34], under nutrient deprivation [35], and in neuromuscular disease [36,37]. In-depth morphological studies are necessary to verify the presence of fused/elongated mitochondria in SMA diaphragm. Preliminary qualitative EM investigations on SMA diaphragms show the presence of mitochondria of variable shape often clustered in shorter columns scattered within the fiber interior (Appendix A) and the lack of “triadic” mitochondria, i.e., mitochondria coupled to the Ca^2+^ Release Units (CRUs) or triads (Appendix A), supporting their misplaced and uneven distribution [38]. The proper “triadic” association is essential for the bi-directional cross-talk between the two organelles, as the Ca^2+^ released from SR during EC coupling enters into the mitochondrial matrix, stimulating ATP production [39,40,41].

Mitochondrial quality control is managed by mitophagy. Mitophagy represents, however, an “extreme decision” for a cell because mitochondria are an essential source of ATP. Moreover, to be degraded by autophagosome, the mitochondria must undergo a prior fragmentation process [42]. It was demonstrated that under starvation, mitochondria protect themselves from mitophagy by promoting fusion in order to maximize ATP production [32]. Only upon prolonged starvation do mitochondria undergo degradation and removal by mitophagy. Chemello and co-workers found a decrease in BNIP3 gene expression and an increase in LC3II in tibialis anterior of muscle-specific Smn1 knockout mice. Moreover, they found a reduction in LAMP1 and LAMP2 and a reduction in autophagic flux, leading them to hypothesize dysfunctional mitochondrial clearance [12]. In accordance with this study, a decrease in gene expression of BNIP3 and the protein level of PARKIN (Figure 6b,c) was found in the SMA diaphragm. However, the general autophagy process seemed to be activated, as indicated by a significant increase in the protein level of LC3BII and by a significant decrease in p62 protein expression (Figure 6a). So, the scenario emerging from the results of our study confirms the hypothesis of a reduction in mitochondria renewal also in diaphragm muscle that, on one hand, could meet the muscle need for the supply of ATP to the cells, but, on the other hand, could lead to an increase in ROS production.

### 3.3. Oxidative Stress in SMA Diaphragm

The presence of redox imbalance and ROS accumulation in SMA mouse diaphragms was revealed by the presence of a great amount of carbonylated proteins (Figure 7a) and by an increase in complexes I and III of the mitochondrial respiratory chain, the main factors responsible for ROS formation (Figure 3) [43]. The increase in ROS production is expected to promote the expression of the antioxidant system composed of NRF2, SOD1, catalase and HSPs, but in the SMA diaphragm, all of these factors were not changed (Figure 7b). The lack of induction of the scavenging system could lead to ROS accumulation. Interestingly, only the expression of mitochondrial antioxidant protein peroxiredoxin 3 (PRDX3), essential for maintaining mitochondrial mass and membrane potential [44], was increased, in accord with the increased TOM20 expression (Figure 7c). Importantly, ROS accumulation could contribute to muscle fatigue, as indicated by an increasing body of evidence [24,45]. Accordingly, the recovery of the contractile performance of the isolated diaphragm after ERGO exposure was found (Figure 8). ERGO is a very water-soluble natural antioxidant molecule accumulating within tissues through the action of a specific organic cation transporter, and it is able to permeate the plasma, placenta and mitochondrial membranes [46,47]. While the exact function of ERGO has yet to be elucidated, it was established as a powerful scavenger mitochondria-derived superoxide species and a protector of mitochondrial constituents from damage by ROS [46,47].

## 4. Materials and Methods

### 4.1. Animals

SMA (SMNΔ7 mice from Jackson Laboratory stock #005025) and control littermate mice 11 days of age were used. This time point was chosen as a compromise to have a higher number of alive sick animals, since the median age of survival of SMA mice is ~13 days. Heterozygous SMNΔ7 mice for SMN1 (SMN+/−) were crossed together to generate offspring with the following genotypes:Homozygous for both transgenes and heterozygous for the null allele (7SMN+/+; SMN2 +/+; SMN+/−—50%). These mice do not show SMA phenotype and are used as breeders.Homozygous for both transgenes and heterozygous for the null allele (7SMN+/+; SMN2 +/+; SMN+/+—25%). These mice do not show SMA phenotype and are used as control mice (WTs).Homozygous for both transgenes and heterozygous for the null allele (7SMN+/+; SMN2 +/+; SMN−/−—25%). These mice show SMA phenotype (SMAs).

In order to achieve statistical significance, the sample size was chosen to use the smallest number of animals; there were no statistical methods employed to predetermine the sample size. The mice were identified through genotyping, but there was no randomization in the experiments. All procedures were approved by the University of Pavia’s Animal Care and Use Committee (protocol reference n° 280/2021-PR) and, in conformity with Italian law, were communicated to the Ministry of Health and local authorities.

### 4.2. Genotyping

A PCR-based assay on tail DNA was used to genotype offspring. The mouse *Smn* knock-out allele was detected using the primers (Sigma Aldrich, St. Louis, MO, USA) in Table 1. The PCR procedure consisted of 5 min of heating at 94 °C, then 40 cycles of 94 °C for 1 min, 53 °C for 1 min, and 72 °C for 1 min, and finally a cycle of 10 min at 72 °C.

### 4.3. Ex Vivo Functional Analysis

The mice were sacrificed by cervical dislocation, the lower part of the chest was removed, and the diaphragm was dissected and immersed in an oxygenated Krebs solution. A stereomicroscope (×10 to ×60) was used to dissect the diaphragm strips (width 1–2 mm) including the central tendon. The diaphragm strips were transferred to the myograph and mounted by hooks between a force transducer (AME801; Aksjeselkapet Mikkroelektronik, Horten, Norway) and a movable shaft used to adjust muscle length. The strips were secured by opening small holes between the ribs and strengthening them with silk thread ligatures [48]. At a constant temperature of 22 °C, the preparations were placed in an organ bath that was filled with Krebs solution bubbled with 95% O_2_ and 5% CO_2._ The preparations were stretched to Lo (the length at which the maximum twitch force is seen), and their electrical stimulation response was evaluated. If a response could be elicited, the preparation was stimulated for ∼30 min with supramaximal, low-frequency (0.03 Hz) stimuli. Time to peak tension (TTP), and time to half relaxation (TTR) were measured. Subsequently, tetanic isometric contractions were evoked (110 Hz, 500 ms, supramaximal amplitude) at Lo. By calculating the decrease in the force of the maximal absolute tetanic force after 20 repeated contractions in a ramp protocol at 0.03, 0.09, 0.3, and 0.9 Hz, the fatigue index was determined [49,50]. The normalized tetanic force was expressed as the maximal tetanic force/muscle cross-sectional area (CSA) (mN/mm^2^), while the fatigue index was expressed as a percentage of the maximal tetanic force. The functional experiments were repeated following incubation for 30 min in Krebs augmented with 0.4% of a water solution containing 20μg of ergothioneine (ERGO), a naturally occurring amino acid derivative of histidine. ERGO has a powerful antioxidant action [51,52], and the dose was chosen based on data present in the literature [53] and modified to allow a translational comparison with human subjects taking ERGO as a dietary supplement.

### 4.4. OXYBLOT Analysis

Frozen diaphragm samples were homogenized at 4 °C in a buffer containing protease inhibitors, β-mercaptoethanol, Tris-HCl 0.5M pH 7.6, NaCl 1M, EDTA 100 mM pH7, and 0.1% NP40. The protein suspension was centrifuged at 13,500 rpm for 20 min at 5 °C. The Oxyblot Protein Oxidation Detection Kit (purchased from Millipore, Vimodrone, Italy) was utilized to detect the protein carbonylation level, using reagents to detect carbonyl groups in the protein side chain. Six micrograms of protein lysate were denatured by adding 12% SDS for a final concentration of 6% SDS. Samples were derivatized, through incubation for 10 min with 2,4-dinitrophenylhydrazine (DNPH), to 2,4-dinitrophenylhydrazone (DNP hydrazone). Polyacrylamide gel electrophoresis was used to separate the DNP-derivatized protein samples (4–20% SDS precast gels, Bio-rad, Hercules, CA, USA), followed by Western blotting. The membranes were stained with Ponceau Red and then incubated with a primary antibody that was specific to the DNP moiety of the proteins, followed by an HRP-antibody conjugate directed against the primary antibody (secondary antibody: goat anti-rabbit IgG). Protein detection was performed using the Amersham ECL Select™ detection system (Cytiva Life Sciences, ex GE Healthcare, Marlborough, MA, USA), which highlights the HPR substrate through a chemiluminescent reaction. Membranes were gained through an analysis software, ImageQuant™ LAS 4000 (GE Healthcare Life Sciences, Milwaukee, WI, USA). Protein oxidation was quantified by defining the oxidative index (OI), i.e., the ratio between densitometric values of the OXYBLOT bands and those stained with Ponceau Red. The OI was expressed relative to control samples to compare different experiments.

### 4.5. Gene Expression Analysis

Total RNA, from diaphragm muscles, was extracted using the Promega SV Total RNA isolation kit; the concentration of RNA was evaluated by using NanoPhotometer N60/N50 (IMPLEN, Westlake Village, CA, USA). A 300 ng amount was reverse-transcribed with Super Script III reverse transcriptase (Invitrogen, Carlsbad, CA, USA) to obtain cDNA. The cDNA was analyzed by real-time PCR with the SyBR Green PCR kit (Applied Biosystem, Foster City, CA, USA), and the data were normalized to tubulin expression (Quantitect primer assay from QIAGEN, Hilden, Germany). Oligonucleotide primers used for real-time PCR are listed in Table 2. To identify genes that expressed differently, the default threshold of 6.0 was utilized. The difference between cycle threshold (Ct) values for each mRNA was calculated by taking the mean Ct of duplicate reactions and subtracting the mean Ct of duplicate reactions for the reference RNA measured on an aliquot from the same RT reaction (CΔt = Ct target gene − Ct reference gene). All samples were then normalized to the ΔCt value of a calibrator sample to obtain a ΔΔCt value (ΔCt target − ΔCt calibrator) (comparative method).

### 4.6. Western Blot Analysis

Diaphragm samples (stored at –80 °C) were pulverized with liquid nitrogen and suspended in a lysis buffer (1% triton × 100, 10% Glycerol, 20 mM TRIS-HCl, 5 mM EDTA, 150 mM NaCl, 100 mM Naf, 2 mM NaPPi, and 1 mM PMSF) supplemented with protease inhibitor cocktail and phosphatase inhibitor cocktail (Sigma-Aldrich, St. Louis, MO, USA). After 20 min on ice, the protein lysate was centrifuged at 18,000× *g* at 4 °C for 20 min, and the supernatant was transferred to a clean Eppendorf tube and stored at –80 °C until it was ready to use. Protein concentration was determined using the RC DCTM protein assay kit (Bio-rad Hercules, CA, USA). Equal amounts of protein sample (40 ug) were loaded on a polyacrylamide gel (Any kD precast gel—Biorad Hercules, CA, USA) and subjected to electrophoresis. Proteins were then electrotransferred to a membrane made of nitrocellulose at 35 mA O/N. After blocking in 5% milk in TBST for 1 h, the membranes were incubated O/N at 4 °C with the specific primary antibody appropriately diluted in a solution of TBST 1X containing 5% BSA or 5% fat-free milk (depending on specificities of antibodies in the datasheet (Table 3)). Lastly, the membranes were then incubated in HRP conjugated secondary antibody, goat-anti-rabbit (1: 10,000, from Millipore, St. Louis, MI. USA) or rabbit-anti-mouse (1:5000, from dako, Santa Clara, CA, USA), diluted in 5% milk, for 1 h at room temperature. The Amersham ECL Select™ detection system (Cytiva Life Sciences, Marlborough, MA, USA, ex GE Healthcare) was utilized to visualize the proteins, which highlights the HPR substrate by a chemiluminescent reaction. The target protein levels were normalized with respect to the amount of a housekeeping protein (tubulin); the phosphorylation levels of some proteins were evaluated by the ratio between phosphorylated and unphosphorylated total forms of the same protein.

### 4.7. Oxygraph-2k for High-Resolution Respirometry (O_2_k HRR)

For the measurements of mitochondrial respiration, the Oxygraph-2k (O_2_k, OROBOROS Instruments, Innsbruck, Austria) was utilized. For each mouse, the diaphragm muscle was properly dissected and divided into two parts in ice-cold BIOPS buffer (biopsy preservation solution: 7.23 mM K_2_EGTA, 5.77 mM Na_2_ATP, 2.77 mM CaK_2_EGTA, 6.56 mM MgCl_2_•6 H_2_O, 20 mM Taurine, 15 mM Na_2_Phosphocreatine, 20 mM Imidazole, 0.5 mM Dithiothreitol (DTT), 50 mM MES, pH 7.1) at 0 °C. Both parts were used in order to perform the analysis in duplicate during the day. Pieces of diaphragm were washed in ice-cold BIOPS for 10 min, permeabilized with 20 µg/mL Saponin (Merck) in BIOPS in agitation at 4 °C, and finally washed with mMiR05 buffer (mitochondrial respiration medium: 3 mM MgCl_2_, 60 mM lactobionic acid 0.5 mM EGTA, 20 mM taurine, 10 mM KH_2_PO_4_, 20 mM 4-(2-hydroxyethyl) piperazine-1-ethanesulfonic acid (HEPES), 110 mM D-sucrose, and 1 g·L-1 bovine serum albumin (pH 7.1)). Muscle samples were transferred into Oroboros-O_2_k chambers; the experiments were performed in Mir05 solution, at 37 °C with an oxygen concentration between 400 and 250 µM to avoid O_2_ limitation of respiration, and the medium was reoxygenated with pure gaseous O_2_ when oxygen concentration decreased below the threshold of 250 µM. Moreover, the measurements were performed in the presence of Blebbistatin (a myosin II-ATPase inhibitor, 25 μM), dissolved in DMSO 5 mM stock [54] in order to avoid spontaneous contraction in the respiration medium of muscle samples. Calibration at air saturation was performed each day before starting experiments. Respiration was determined using substrate-uncoupler-inhibitor titration (SUIT) protocols previously described [55,56], with modifications. Glutamate and malate (10 mM and 4 mM, respectively) were added to measure non-phosphorylating resting mitochondrial respiration in the absence of adenylates, in order to evaluate the consumption of O_2,_ mainly determined by the leakage of protons through the inner mitochondrial membrane (“LEAK” respiration). Succinate (10 mM) was then used to support convergent electron flow into the Q-junction through Complexes I and II (determining the LEAK of both Complexes I and II). ADP was added until reaching a 10 mM final saturating concentration, to obtain maximal ADP-stimulated mitochondrial respiration (OXPHOS capacity). Titrations with FCCP, the uncoupler protonophore carbonylcyanide-p trifluoromethoxyphenylhydrazone (a few steps of 1 µM), were performed to determine electron transfer system (ETS) capacity. Rotenone (Rot, 0.5 µM added to inhibit Complex I) and Antimycin A (AmA, 2.5 µM to inhibit Complex III and thus the mitochondrial respiratory chain) were added for the determination of maximal respiratory uncoupled efficiency and residual oxygen consumption (ROX) independently by mitochondria, respectively. Prior to AmA, Cytochrome C (10 µM) was added to the chambers to evaluate the outer mitochondrial membrane integrity: an increase in oxygen flux of more than 15% would indicate damaged organelles. At the end of each experiment, muscle samples were taken from the chambers, washed in PBS, and centrifuged for 10 min at 14,000× *g* at 4 °C, and frozen in liquid nitrogen to be stored at −80 °C until further determinations. All of the mitochondrial respiration parameters analyzed were corrected for O_2_ flux resulting from residual O_2_ consumption (ROX) and were normalized by the value of citrate synthase activity (see below) [57].

### 4.8. Citrate Synthase Activity

CS activity was assessed using the Citrate Synthase Activity Assay kit (Merck, Darmstadt, Germany). Muscle samples were homogenized in CS Assay Buffer provided by the kit and centrifuged at 10,000× *g* for 20 min at 4 °C after being kept on ice for 40 min. RC-DC™ protein assay (Bio-rad Hercules, CA, USA) was used to determine protein concentration; 5 µg of protein lysate was used to test CS activity. The absorbance was measured at 412 nm at the initial time (Tinitial), after adding the appropriate Reaction Mix, with a plate reader (CLARIOstar Plus, BMG Labtech, Ortenberg, Germany), and then the measurement was followed for 45 min, taking an absorbance measurement (A412) every 5 min. CS enzyme activity was calculated by interpolating the value of the final time (Tfinal) on a standard curve whose points was scalar concentrations of a solution of known concentration (2 nmol/µL) of GSH standard solution. Values for citrate synthase activity were used to normalize the parameters evaluated by using O_2_k HRR.

### 4.9. Electron Microscopy (EM)

Intact diaphragms were fixed at room temperature with 3.5% glutaraldehyde in 0.1M Na Cacodylate buffer, pH 7.2 for several hours. Small pieces of fixed diaphragms were then processed, cut, and stained as in Boncompagni et al. 2009 [38]. Sections were viewed and photographed in a 120 kV JEM-1400 Flash Transmission Electron Microscope (Jeol Ltd., Tokyo, Japan) equipped with CMOS camera (Matataki and TEM Center software Ver. 1.7.22.2684 (Jeol Ltd., Tokyo, Japan)).

### 4.10. Statistical Analysis

The data were expressed as mean ± S.D. The statistical significance of the differences between the averages was determined with the Student t-test; a probability of less than 5% (*p* < 0.05) was considered significant. The statistical analysis was performed using GraphPad Prism 9.0 software. The sample size was predetermined based on the published literature and chosen to use the fewest number of animals to achieve statistical significance.

## 5. Conclusions

Current pharmacological and cellular approaches, mainly focusing on motor neuron alterations, are only partially effective in ameliorating SMA clinical presentation. To achieve new and real expectations for SMA disease treatment in the future, it is necessary to develop alternative and parallel sites of intervention, especially for the patients who unfortunately already are not treatable with the newly discovered drugs (Spinraza, Risdiplam, and AVXS-101 (Zolgensma)). In particular, the improvement of diaphragm function could be useful to improve the quality of life of all patients with SMA.

The results obtained in the present study give the first demonstration of an intrinsic energetic imbalance and a significant ROS accumulation not scavenged by the antioxidant enzyme pool in the diaphragm of SMNΔ7 mice. Both conditions can contribute to muscle fatigue and, in the long run, lead to diaphragm wasting and respiratory failure. The significant increase in fatigue resistance of isolated diaphragm from SMA mice following ERGO exposure supports the rationale to test the effect of dietary supplementation with ERGO in SMA mice to improve their life quality and increase their lifespan.

## Figures and Tables

**Figure 1 ijms-24-14953-f001:**
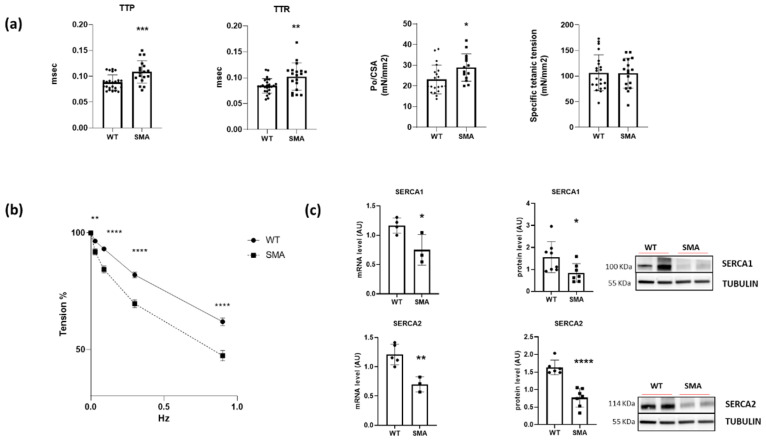
Alterations of contractile functions in SMA diaphragm. (**a**) Mean values of twitch time to peak (TTP), twitch half relaxation time (TTR), twitch specific tension (Po/CSA), and specific tetanic tension determined in ex vivo functional analysis in the diaphragm muscles (WT, n = 21; SMA, n = 17). (**b**) Fatigue index (percentage of the maximal tetanic force) (WT, n = 12; SMA, n = 12). (**c**) Gene (by RT-PCR) expression (WT, n = 5; SMA, n = 3) and protein (by Western blot) expression (WT, n = 8; SMA, n = 7) of sarcoplasmic reticulum Ca^2+^ ATPase pumps (SERCA1 and 2). The level of protein target was normalized against the level of the housekeeping tubulin measured in the same blot. Representative Western blots are shown. Bars represent means± SD. Individual data are represented as scatter plots. * *p* ≤ 0.05 ** *p* ≤ 0.01 *** *p* ≤ 0.001**** *p* ≤ 0.0005.

**Figure 2 ijms-24-14953-f002:**
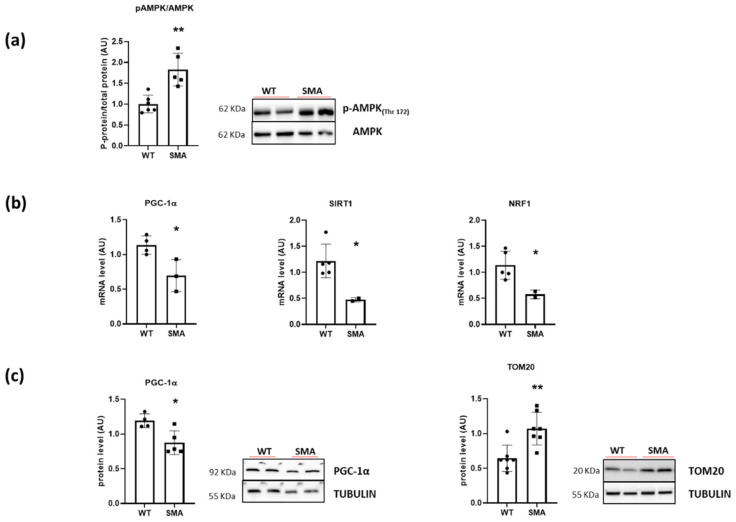
Energy imbalance in SMA diaphragm. (**a**) Mean values of the ratio between the content in the phosphorylated (p) and total forms for AMPK determined by Western blots (WT, n = 6; SMA, n = 5). (**b**) Gene expression of PGC1α, Sirtuin1 (SIR1) and NRF1 by RT-PCR (WT, n = 5; SMA, n = 3). (**c**) Protein expression of PGC1α (WT, n = 4; SMA, n = 5) and mitochondrial import receptor subunit TOM-20 (WT, n = 7; SMA, n = 7) determined by Western blots. The level of protein target was normalized against the level of the housekeeping tubulin measured in the same blot. Representative Western blots are shown. Bars represent means ± SD. Individual data are represented as scatter plots. * *p* ≤ 0.05 ** *p* ≤ 0.01.

**Figure 3 ijms-24-14953-f003:**
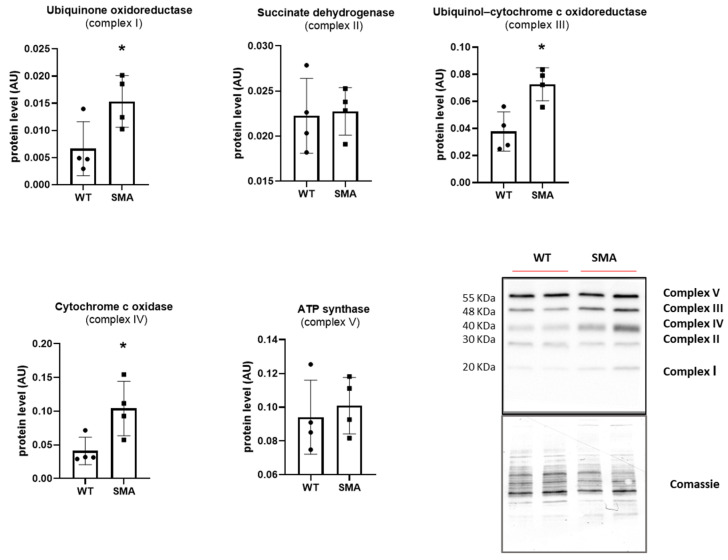
Up-regulation of OXPHOS complexes in SMA diaphragm. Experiments were performed on controls (WT, n = 4) and SMA phenotype (SMA, n = 4) diaphragms. Mean values of the components of the mitochondrial respiratory chain determined by Western blots. The level of protein target was normalized against the level of the same blot. stained with Comassie. Representative Western blots are shown. Bars represent means ± SD. Individual data are represented as scatter plots. * *p* ≤ 0.05.

**Figure 4 ijms-24-14953-f004:**
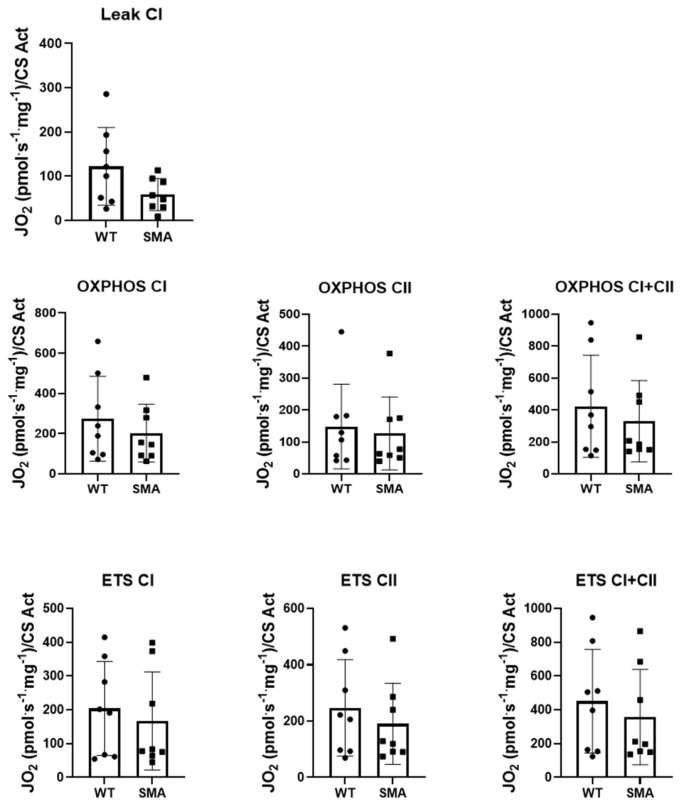
Mitochondrial respiration indices (LEAK—Resting non-phosphorylating electron transfer, OXPHOS—Oxidative phosphorylation, ETS—electron transfer system) are not changed in the complexes evaluated (Complex I—CI and Complex II—CII, both complexes—CI + CII) in SMA diaphragm. Experiments were performed on controls (WT, n = 8) and SMA phenotype (SMA, n = 8) diaphragms. Mean values of the mitochondrial respiration indices. Bars represent means ± SD. Individual data are represented as scatter plots.

**Figure 5 ijms-24-14953-f005:**
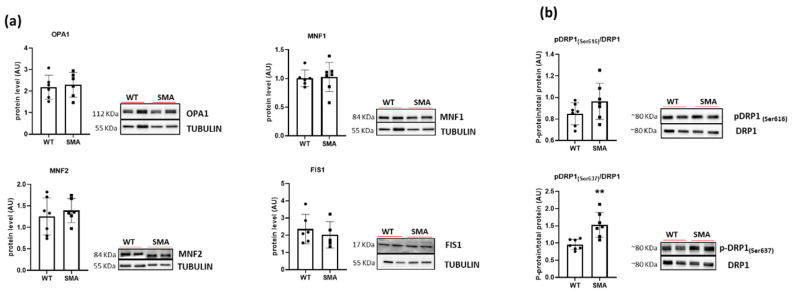
Alteration of markers of mitochondrial dynamics in SMA diaphragm. (**a**) Protein levels of OPA1, MNF1, MNF2 and FIS1 determined by Western blots (WT, n = 6; SMA, n = 6). (**b**) Phosphorylation of serine 616 and serine 637 of DPR1 (WT, n = 7; SMA, n = 7). The level of protein target was normalized against the level of the housekeeping tubulin measured in the same blot. Representative Western blots are shown. Bars represent means ± SD. Individual data are represented as scatter plots. ** *p* ≤ 0.01.

**Figure 6 ijms-24-14953-f006:**
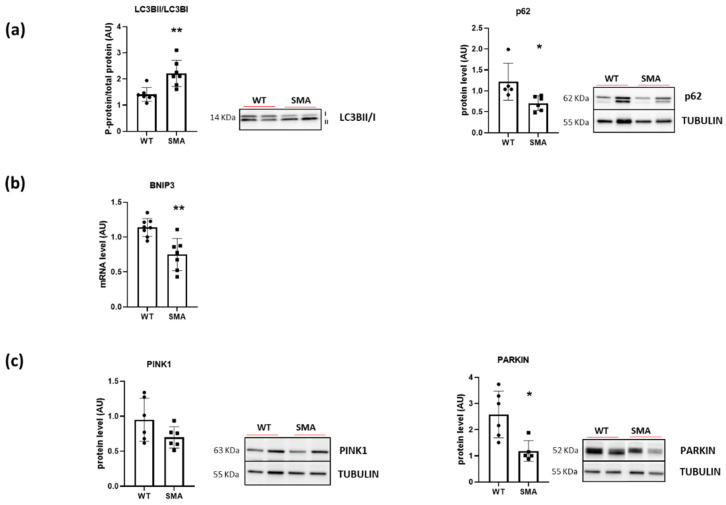
Alteration in autophagy and mitophagy processes in SMA diaphragm. (**a**) The ratio between the active form and the inactive form of LC3B (LC3BII/LC3BI) (WT, n = 7; SMA, n = 7) and protein levels of p62 (WT, n = 5; SMA, n = 6), determined by Western blots. (**b**) Gene expression of BNIP3 by RT-PCR (WT, n= 8; SMA, n= 7). (**c**) Protein levels of mitophagy makers PINK1 and PARKIN (WT, n = 6; SMA, n = 6), determined by Western blots. The level of protein target was normalized against the level of the housekeeping tubulin measured in the same blot. Representative Western blots are shown. Bars represent means ± SD. Individual data are represented as scatter plots. * *p* ≤ 0.05 ** *p* ≤ 0.01.

**Figure 7 ijms-24-14953-f007:**
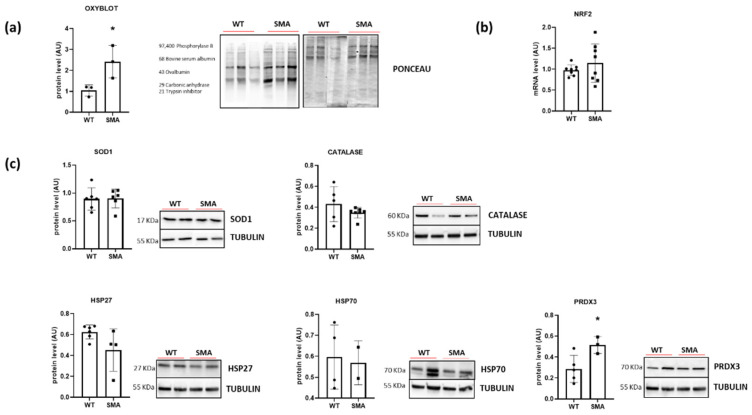
Presence of redox imbalance and ROS accumulation in SMA diaphragm. (**a**) Protein levels of carbonylated proteins revealed by Oxyblot analysis (WT, n= 3; SMA, n= 3). (**b**) Gene expression of NRF2 by RT-PCR (WT, n = 8; SMA n = 7). (**c**) Protein levels of SOD1 (WT, n = 6; SMA, n = 6), Catalase (WT, n = 5; SMA, n = 7), HSP27 (WT, n = 6; SMA, n= 4), HSP70 (WT, n = 4; SMA, n = 2) and Piroxiredoxin 3 (PRDX3) (WT, n= 5; SMA, n= 3), determined by Western blots. The level of protein target was normalized against the level of the housekeeping tubulin measured in the same blot. Representative Western blots are shown. Bars represent means ± SD. Individual data are represented as scatter plots. * *p* ≤ 0.05.

**Figure 8 ijms-24-14953-f008:**
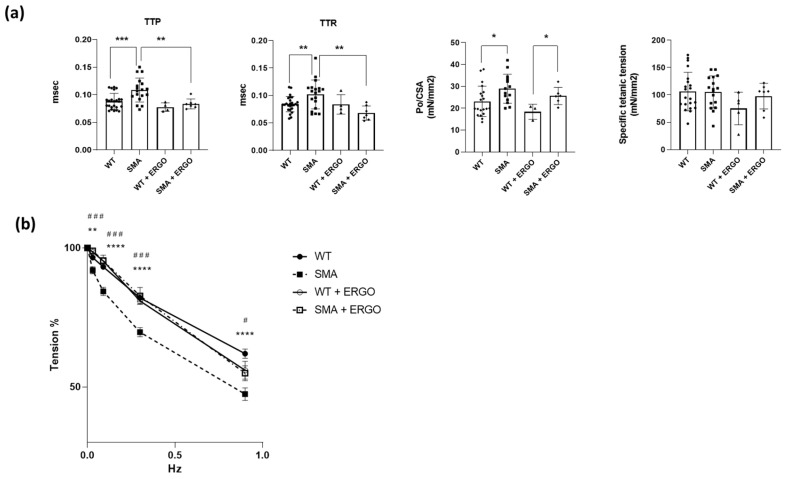
Recovery of contractile functions in SMA diaphragm following ergothioneine exposure. Experiments were performed on diaphragms of controls (WT), SMA phenotype (SMA), controls treated with ergothioneine (WT + ERGO), and SMA phenotype treated with ergothioneine (SMA + ERGO). (**a**) Mean values of twitch time to peak (TTP), twitch half relaxation time (TTR), twitch specific tension (Po/CSA) and specific tetanic tension (WT, n = 21; SMA, n = 17; WT + ERGO, n = 5 and SMA + ERGO, n = 7). (**b**) Fatigue index (percentage of the maximal tetanic force) (WT, n = 12; SMA, n = 12, WT + ERGO, n = 3 and SMA + ERGO, n = 8). Bars represent means ± SD. Individual data are represented as scatter plots. * *p* ≤ 0.05 ** *p* ≤ 0.01 *** *p* ≤ 0.001 **** *p* ≤ 0.0005 SMA vs. WT; # *p* ≤ 0.05 ### *p* ≤ 0.001 SMA vs. SMA+ERGO.

**Table 1 ijms-24-14953-t001:** Primers used for PCR assay to genotype offspring.

*SMN1* WT FORWARD	5′CTCCGGGATATTGGGATTG 3′
*SMN1* WT REVERSE	5′ TTTCTTCTGGCTGTGCCTTT 3′
*SMN1* MUTANT REVERSE	5′ GGTAACGCCAGGGTTTTCC 3′
*SMN2* WT FORWARD	5′ CTGACCTACCAGGGATGAGG 3′
*SMN2* TRANSGENE	5′ GGTCTGTTCTACAGCCACAGC 3′
*SMN2* WT REVERSE	5′CCCAGGTGGTTTATAGACTCAGA 3′
*SMN*Δ7 TRANSGENE 01	5′ TCCATTTCCTTCTGGACCAC 3′
*SMN*Δ7 TRANSGENE 02	5′ ACCCATTCCACTTCCTTTTT 3′
*SMN*Δ7 POSITIVE CTRL FORWARD	5′ CAAATGTTGCTTGTCTGGTG 3′
*SMN*Δ7 POSITIVE CTRL REVERSE	5′ GTCAGTCGAGTGCACAGTTT 3′

**Table 2 ijms-24-14953-t002:** Oligonucleotide primers used for real-time PCR.

	Forward Primer	Reverse Primer
*SERCA1*	TGTTTGTCCTATTTCGGGGTG	AATCCGCACAAGCAGGTCTTC
*SERCA2*	GAGAACGCTCACACAAAGACC	CAATTCGTTGGAGCCCCAT
*PGC1α*	ACCCCAGAGTCACCAAATGA	CGAAGCCTTGAAAGGGTTATC
*SIRT1*	CCGCGGATAGGTCCATATACT	AACAATCTGCCACAGCGTCA
*NRF1*	GCACCTTTGGAGAATGTGGT	CTGAGCCTGGGTCATTTTGT
*NRF2*	TTCTTTCAGCAGCATCCTCTCCAC	ACAGCCTTCAATAGTCCCGTCCAG
*BNIP3*	TTCCACTAGCACCTTCTGATGA	GAACACCGCATTTACAGAACAA

**Table 3 ijms-24-14953-t003:** Antibodies used for Western Blot Analysis.

Primary Antibody	Species	Dilution	Supplier	Catalog Number
p-AMPK	Rabbit	1:1000	Cell Signaling	#4188
AMPK	Rabbit	1:1000	Cell Signaling	#2532
PGC1α	Rabbit	1:1000	Abcam	Ab54481
TOM20	Rabbit	1:1000	Santa Cruz	sc-11415
OPA1	Mouse	1:3000	Abcam	Ab157457
MNF1	Mouse	1:1000	Abcam	Ab57602
MNF2	Rabbit	1:1000	Abcam	Ab50843
FIS1	Rabbit	1:1000	Abcam	Ab71498
DRP1 pSer616	Rabbit	1:1000	Cell Signaling	#3455
DRP1 pSer637	Rabbit	1:1000	Cell Signaling	#4867
DRP1	Rabbit	1:1000	Cell Signaling	#8570
SOD1	Rabbit	1:1000	Abcam	Ab16831
Catalase	Rabbit	1:1000	Abcam	Ab52477
Hsp27	Mouse	1:1000	Abcam	Ab2790
Hsp70	Mouse	1:1000	Abcam	Ab47455
PRDX3	Rabbit	1:1000	Abcam	Ab73349
LC3B	Rabbit	1:1000	Sigma	L7543
p62	Rabbit	1:2000	Cell Signaling	#5114
PINK1	Rabbit	1:1000	Invitrogen	#PA1-16604
PARKIN	Mouse	1:2000	Invitrogen	#39-0900
Tubulin	Mouse	1:2000	Sigma	T6199
OXPHOS	Mouse	1:1000	Abcam	Ab110413
SERCA1	Rabbit	1:1000	Cell Signaling	#12293
SERCA2	Rabbit	1:1000	Cell Signaling	#9580

## Data Availability

Not applicable.

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
