# Peer review of "Diaphragm Fatigue in SMNΔ7 Mice and Its Molecular Determinants: An Underestimated Issue"

_ijms, 2023, doi:10.3390/ijms241914953_

Round 1

Reviewer 1 Report

Overall, this manuscript is of high quality and importance to the SMA community. There are very minor changes recommended. 

- On line 120, please qualify the word "respiration" with the word "cellular" to provide better context. 

- In Fig 6c - the WB depicted for PARKIN appears that the top panel (PARKIN) has a cut between WT and SMA, but a similar cut is not present in the bottom panel (TUBULIN). Please verify that the lanes for the top and bottom panels are the same. 

- In many graphs, particularly those with p<0.05 (single asterisk) it is difficult to differentiate the asterisk from a potential data point. Placing all asterisks near the top of the graph rather than just atop the bar will remedy this. 

- The depicted example WBs where total protein is shown (Fig 3 Coomassie; Fig 7 Ponceau) appear overly white balanced. Please provide the original versions. 

- The methods sections is excellently written with one small exception, please include the product / catalog numbers for the antibodies in Table 3. Simply listing the supplier does not provide enough information to the reader, given that these suppliers sell multiple antibodies for the same protein. 

The quality of English language is mostly good in this manuscript.  There are a few places, primarily in the abstract and the introduction that could use some editing to improve readability. 

Author Response

Dear Editor and Reviewer, many thanks for the time dedicated to revising our manuscript. We modified the text according to the comments and suggestions provided. New or modified sentences are highlighted in the revised version of the text. Line numbers are referred to the revised manuscript.

Overall, this manuscript is of high quality and importance to the SMA community. There are very minor changes recommended.

We thank the Reviewer for dedicating time to revising our manuscript. We are happy to know that our research findings raised the reviewer’s interest.

 - On line 120, please qualify the word "respiration" with the word "cellular" to provide better context. 

Done

- In Fig 6c - the WB depicted for PARKIN appears that the top panel (PARKIN) has a cut between WT and SMA, but a similar cut is not present in the bottom panel (TUBULIN). Please verify that the lanes for the top and bottom panels are the same. 

Actually, the tubulin panel is cut in the same position and the same samples are compared. The Reviewer can verify the correctness of the blots by viewing the file “WB paper 2023” sent to the Editor.

- In many graphs, particularly those with p<0.05 (single asterisk) it is difficult to differentiate the asterisk from a potential data point. Placing all asterisks near the top of the graph rather than just atop the bar will remedy this. 

We try to solve the problem by moving the asterisks higher on the bars and zooming them. We hope the action could be enough to not confuse the asterisk with the dots.

- The depicted example WBs where total protein is shown (Fig 3 Coomassie; Fig 7 Ponceau) appear overly white balanced. Please provide the original versions. 

The WB images are the original versions as the Reviwer can see in the file “WB paper 2023” sent to the Editor. The Fig. 3 (Comassie) is the original one, whereas in Fig. 7 (Ponceau) the problem arose from the inclusion during the blot analysis of the darkest band of the marker. We solved the problem by redoing the analysis. We replaced the WB images in the paper.

- The methods sections is excellently written with one small exception, please include the product / catalog numbers for the antibodies in Table 3. Simply listing the supplier does not provide enough information to the reader, given that these suppliers sell multiple antibodies for the same protein. 

Done

The quality of English language is mostly good in this manuscript.  There are a few places, primarily in the abstract and the introduction that could use some editing to improve readability. 

We thank the Reviwer for the suggestion. We have made corrections to the English language in all the text.

Reviewer 2 Report

Spinal muscular atrophy is an autosomal recessive neuromuscular disease affecting low motor neurons. The SMN1 gene deletion leads to death of α–motor neurons. One of the earliest events detected is the defects at the NMJ in mouse model of SMA. The defects are followed by loss of NMJ, denervation of the muscle with subsequent weakness and atrophy. Nevertheless, the weakness and fatigability of diaphragm muscle had been noticed, but the mechanism is rarely been investigated. Here the study investigated the contractility and the associated mitochondrial function complex of the diaphragm muscles from SMA mice. According to the proposed mechanism, ergothioneine (ERGO) has been tested and demonstrated the effective recue on the diaphragm fatigability.

Basically, the investigation is dedicated tackling with the functional disturbance as well as the molecular mechanisms on the respiratory dysfunction in SMA mouse model. The experiments are appropriately designed. The manuscript is well written and has clearly described the underlying mechanism leading to the muscle weakness in the mice.  

I only have a few minor concerns which are listed as follows:

1.      Line 46, “in locomotor muscle of patients….” needs to be changed into “in the locomotor muscles of patients…..”.  

2.      Two sentences between line 51 and 55 need to be rephrased.

3.      Line 59, “molecular level the human SMA diaphragm” needs to be changed into “molecular level of the human SMA diaphragm”.

4.      Line 59, is “NMJs” the abbreviation of “neuromuscular junctions”??

5.      The sentence between line 64 and 68 is too long to be understood, which needs to be rephrased.

6.      Line 100, “analysis in WT(n=21) …..” needs to be changed into “analysis in the diaphragm muscles in WT (n=21)….”.

7.      Line 103, “Ca2+ ATPase (SERCA1 and 2) pumps” needs to be changed into “Ca2+ ATPase pumps (SERCA1 and 2)”.

8.      Line 106, What does the “****” mean??

9.      A comma needs to be used to separate the mice strain and the numbers of experiment animals.

10.  The Western blotting image in Fig. 2C shows the variable expression of PGC-1α in the SMA mice, although the count protein level in the mice is decreased. It needs to be explained.

11.  Line 121, “sustained by complex I, complex I……..” should be changed into “sustained by complex I, complex II………”.

12.  In figure legend of figure 4, the abbreviations for the “Leak”, “OXPHOS”, “CI”, “CII”, and “ETS” need to be written.

13.  In figure 5a, a few typos: “MFN1” should be “MNF1”, similarly “MFN2” should be “MNF2”.

14.  In figure 7, there are only two animals (SMA n = 2) to evaluate the HSP70 expression. A few other experiments also employed only 3 animals. It might not be adequate for the comparison in statistical analysis.

15.  The first paragraph in Discussion section can move to the “Introduction” or be left out.

16.  The long sentence between 210 and 212 needs to be rephrased.

17.  Line 216, what is the meaning of “NIV”??

18.  The sentence between line 223 and 225 needs to be rephrased.

19.  Line 228, “Ca2+” needs to be changed to “Ca2+”.

20.  In section 3.2, The “ATP availably” may be changed to “the available ATP”.

21.  Line 257, “a cell culture SMN depletion model” needs to be changed to “a SMN depletion cell model”.

22.  Line 281, “Ca2+” needs to be changed to “Ca2+”.

23.  The sentence between line 310 and 312 needs to be rephrased.

24.  Line 364, “O2” and “CO2” need to be changed to “O2” and “CO2

25.  Line 373, “mN/mm2” needs to be changed to “mN/mm2”.

26.  Line 379, it needs a full stop at the end of sentence.

27.  Line 386, “6µg of protein lysate….” needs to be changed to “Six microgram of protein lysate….”.

28.  Line 403, does the “skeletal samples” mean the “diaphragm muscles”??

29.  The long sentence between line 403 and 405 needs to be rephrased.

30.  Line 414, the “ΔCttarget” should be “ΔCt target” and “ΔCtcalibrator” be “ΔCt calibrator”.

31.  Two sentences between line 425~427 need to be rephrased.

32.  Line 431, the “HRPconjugated” should be “HRP conjugated”.

33.  There are a few numbers in Section 4.7, needs to be presented with the “low case” form such as “MgCl2•6 H2O” should be “MgCl2•6 H2O”.

34.  The long sentence between line 466 and 469 needs to be rephrased.

The quality of English language is fine. 

Author Response

Dear Editor and Reviewer, many thanks for the time dedicated to revising our manuscript. We modified the text according to the comments and suggestions provided. New or modified sentences are highlighted in the revised version of the text. Line numbers are referred to the revised manuscript.

Basically, the investigation is dedicated tackling with the functional disturbance as well as the molecular mechanisms on the respiratory dysfunction in SMA mouse model. The experiments are appropriately designed. The manuscript is well written and has clearly described the underlying mechanism leading to the muscle weakness in the mice.

We thank the Reviewer for dedicating time in revising our manuscript. We are happy to know that our research findings raised reviewer’s interest

  1. Line 46, “in locomotor muscle of patients….” needs to be changed into “in the locomotor muscles of patient s…..”.  

Done

  1. Two sentences between line 51 and 55 need to be rephrased.

Done

  1. Line 59, “molecular level the human SMA diaphragm” needs to be changed into “molecular level of the human SMA diaphragm”.

Done

  1. Line 59, is “NMJs” the abbreviation of “neuromuscular junctions”??

We have explained the meaning of the abbreviation

  1. The sentence between line 64 and 68 is too long to be understood, which needs to be rephrased.

Done

  1. Line 100, “analysis in WT(n=21) …..” needs to be changed into “analysis in the diaphragm muscles in WT (n=21)….”.

Done

  1. Line 103, “Ca2+ ATPase (SERCA1 and 2) pumps” needs to be changed into “Ca2+ATPase pumps (SERCA1 and 2)”.

Done

  1. Line 106, What does the “****” mean??

We have inserted the explanation of what it means in the legend

  1. A comma needs to be used to separate the mice strain and the numbers of experiment animals.

Done

  1. The Western blotting image in Fig. 2C shows the variable expression of PGC-1α in the SMA mice, although the count protein level in the mice is decreased. It needs to be explained.

The WB image we included in the paper is one of many obtained. Actually, the samples showed some variability but overall the total expression of PGC1α was decreased in the SMA mice. We replaced the image with a more representative one. 

  1. Line 121, “sustained by complex I, complex I……..” should be changed into “sustained by complex I, complex II………”.

Done

  1. In figure legend of figure 4, the abbreviations for the “Leak”, “OXPHOS”, “CI”, “CII”, and “ETS” need to be written.

Done

  1. In figure 5a, a few typos: “MFN1” should be “MNF1”, similarly “MFN2” should be “MNF2”.

Done

  1. In figure 7, there are only two animals (SMA n = 2) to evaluate the HSP70 expression. A few other experiments also employed only 3 animals. It might not be adequate for the comparison in statistical analysis.

The Reviewer is right. This analysis was done at a later time and unfortunately, there were only two samples left. In this case, statistical analysis was not performed.

  1. The first paragraph in Discussion section can move to the “Introduction” or be left out.

The paragraph in Discussion section was left out.

  1. The long sentence between 210 and 212 needs to be rephrased.

This paragraph was left out.

  1. Line 216, what is the meaning of “NIV”??

This paragraph was left out.

  1. The sentence between line 223 and 225 needs to be rephrased.

Done

  1. Line 228, “Ca2+” needs to be changed to “Ca2+”.

Done

  1. In section 3.2, The “ATP availably” may be changed to “the available ATP”.

Done

  1. Line 257, “a cell culture SMN depletion model” needs to be changed to “a SMN depletion cell model”.

Done

  1. Line 281, “Ca2+” needs to be changed to “Ca2+”.

Done

  1. The sentence between line 310 and 312 needs to be rephrased.

Done

  1. Line 364, “O2” and “CO2” need to be changed to “O2” and “CO2

Done

  1. Line 373, “mN/mm2” needs to be changed to “mN/mm2”.

Done

  1. Line 379, it needs a full stop at the end of sentence.

Done

  1. Line 386, “6µg of protein lysate….” needs to be changed to “Six microgram of protein lysate….”.

Done

  1. Line 403, does the “skeletal samples” mean the “diaphragm muscles”??

We have replaced “skeletal samples” with “diaphragm muscles”

  1. The long sentence between line 403 and 405 needs to be rephrased.

Done

  1. Line 414, the “ΔCttarget” should be “ΔCt target” and “ΔCtcalibrator” be “ΔCt calibrator”.

Done

  1. Two sentences between line 425~427 need to be rephrased.

Done

  1. Line 431, the “HRPconjugated” should be “HRP conjugated”.

Done

  1. There are a few numbers in Section 4.7, needs to be presented with the “low case” form such as “MgCl2•6 H2O” should be “MgCl2•6 H2O”.

Done

  1. The long sentence between line 466 and 469 needs to be rephrased.

Done